# Misjudgement of One’s Own Performance? Exploring Attention Deficit (Hyperactivity) Disorder (ADHD) and Individual Difference in Complex Music and Foreign Language Perception

**DOI:** 10.3390/ijerph20196841

**Published:** 2023-09-27

**Authors:** Christine Groß, Valdis Bernhofs, Eva Möhler, Markus Christiner

**Affiliations:** 1Jazeps Vitols Latvian Academy of Music, K. Barona Street 1, LV-1050 Riga, Latvia; christine.michaela.gross@jvlma.lv (C.G.); valdis.bernhofs@jvlma.lv (V.B.); 2Department of Child and Adolescent Psychiatry, Saarland University Hospital, G-66421 Homburg, Germany; eva.moehler@uks.eu; 3Centre for Systematic Musicology, Faculty of Arts and Humanities, University of Graz, Glacisstraße 27, A-8010 Graz, Austria

**Keywords:** ADHD, complex stimuli processing, musicality, foreign language perception, short-term memory, self-evaluation, speech perception deficit, self-assessment

## Abstract

In previous research, we detected that children and adolescents who were diagnosed with ADHD showed deficits in both complex auditory processing of musical stimuli and in musical performance when compared to controls. In this study, we were interested in whether we could detect similar or distinct findings when we use foreign speech perception tasks. Therefore, we recruited musically naïve participants (*n* = 25), music-educated participants (*n* = 25) and participants diagnosed with ADHD (*n* = 25) who were assessed for their short-term memory (STM) capacity and the ability to discriminate music and speech stimuli and we collected self-ratings of the participants’ music and language performances. As expected, we found that young adults with ADHD show deficits in the perception of complex music and difficult speech perception stimuli. We also found that STM capacity was not impaired in young adults with ADHD and may not persist into young adulthood. In addition, subjective self-estimation about the participants’ language and music performances revealed that the ADHD group overestimated their performance competence relatively compared to both control groups. As a result, the findings of our study suggest that individuals diagnosed with ADHD require a special training program that not only focuses on improving performance in perceptual skills of music and language but also requires metacognitive training to develop realistic self-assessment skills.

## 1. Introduction

Attention deficit hyperactivity disorder (ADHD) is associated with symptoms such as hyperactivity, impulsivity and/or inattention. Following Polanczyk and colleagues [1], 5–10% of children and adolescents are affected by ADHD and a substantial percentage of around 60% of children remain affected into adulthood [2]. Studies have shown that individuals diagnosed with ADHD show lower academic achievements and educational outcomes in various domains [3]. This affects a number of different capacities such as academic achievement [4,5,6], arithmetic ability [7], reading and comprehension skills [8,9,10,11], the ability to attend to speech in typically noisy listening environments [12] and working memory capacity [7,11,13,14,15,16,17]. Mixed results were reported for deficits of the phonological short-term memory (STM) which more likely seem to depend on age. Studies on adults diagnosed with ADHD and the phonological STM reported small to moderate deficits [13], while studies on children reported the opposite [18,19]. This may be related to the fact that some deficits such as working memory capacity persist into adulthood, while others do not [13,20]. However, despite the fact that individuals diagnosed with ADHD show poorer performance in many domains, they tend to overestimate and misinterpret their own performance [21], which stands in opposition to musicality research where self-estimations of individuals were highly intercorrelated [22,23]. The discrepancy between self-perceptions and performance of individuals with ADHD has also been referred to as the positive illusory bias [21,24]. Research has suggested that ADHDs’ overrating protects their self-image [25]. The fact that individuals diagnosed with ADHD perform lower in many cognitive capacities as well as the fact that they tend to overestimate their performance shows the serious demand for understanding the impact of ADHD on cognitive abilities and self-recognition in more detail.

Generally speaking, individuals affected by ADHD show several impairments which also affect musical performance [26,27]. Adolescents with ADHD showed inferior rhythmical and pitch improvisation performance in music compared to their non-affected peers [26]. It can be suggested that lower musical performance may partly be related to motor deficits and sensorimotor integration impairments. Both deficits have been reported to be typical impairments of individuals showing signs of ADHD [28,29]. Individuals with elaborate music and singing skills represent the opposite and possess enhanced motor and sensorimotor skills which may be a fundamental reason why individuals possessing high musical ability also possess enhanced language skills [30,31,32]. Enhanced motor and sensorimotor skills are particularly believed to be key elements in explaining superior generational music (playing musical instruments or singing) and language performances (e.g., foreign language pronunciation) [31,32,33].

Individuals diagnosed with ADHD also show neurodevelopmental deficits in the auditory cortex, affecting processing of auditory stimuli. The auditory cortex is connected to various prefrontal and parietal brain regions through cortical and subcortical networks and also controls sensorimotor [34,35,36], cognitive [37] and language-related information [38,39] but also includes attentional networks [27,40,41]. Neurophysiological research has provided evidence that individuals diagnosed with ADHD show pronounced asynchrony of the cortical auditory-evoked response complex (P1-N1-P2 complex) which is a crucial predictor explaining individual differences in sound perception [27,41,42].

Neurodevelopmental deficits may be one reason why individuals diagnosed with ADHD also show perceptual deficits in multiple musical domains [26,27,41] leading to lower performance in perception of temporal auditory information such as duration discrimination, timing skills and the ability to perceive rhythmic information [43,44,45,46,47]. Children diagnosed with ADHD show higher-order auditory processing deficits which lowers performance in discriminating changes in paired rhythmic and melodic musical statements [27]. They also seem to be unable to generate an internal beat when listening to rhythmic sounds [48] and suffer from fundamental hearing difficulties [49]. Auditory deficits may also be a major reason why individuals diagnosed with ADHD seem to have deficits in music and language.

Music and language are both acoustic phenomena which require a precise ability to discriminate auditory information. Therefore, overlaps between music and language are rather salient when foreign languages are learnt [22,50,51,52]. In initial foreign-language-learning settings, linguistic content is highly reduced and learners rely on incorporating acoustic information which resembles the learning of new melodies [53]. It is therefore not surprising that positive transfer from music to language has in particular been observed for foreign language capacity in which new, unfamiliar utterances or non-native contrasts are acquired [33,50,51,52,54,55,56,57,58]. Research has shown that musical ability and musical training are associated with enhanced speech perception [59], with improved foreign speech segmentation [22,60] and foreign language aptitude [22]. However, enhanced intelligibility of newly learnt utterances is also associated with musical aptitude measures [61] and tone frequency linked to the acquisition of tone syllable learning in Mandarin [33]. Subsequently, there is a common agreement that enhanced music perception also positively influences language perception. As musical and language ability are intertwined, it could be suggested that individuals with ADHD show not only deficits in discriminating complex musical stimuli but also in discriminating new unfamiliar foreign languages.

In the context of this study, the term “complex” refers to difficult music discrimination tasks that consist of longer melodies which have to be remembered and to difficult language measures that consist of more than one language constituent which have to be remembered and discriminated. In this study, the term “unfamiliarity” means that individuals do not speak, comprehend or have not learnt any of the languages.

Research on initial foreign language acquisition processes and ADHD has surprisingly been ignored as far as we know, although this is a crucial research direction since foreign language learning is a situation everyone experiences when entering school.

Measuring musical ability and measuring initial foreign language capacity also have a lot in common. Tests are often based on the idea that stimuli are less familiar, rather new or even unfamiliar to the participants. For instance, the language ability measure LLAMA uses a seldom-spoken British Columbian Indian language as stimuli to minimize educational differences [62]. Likewise, musical ability measures most often consist of unfamiliar short or long sequences of paired musical statements in which deviations from the first statement have to be detected and depend on the task defined [63,64,65].

A number of tests have been developed to assess musical ability. Early developed musicality tests include the Seashore Test [66,67], the Wing Standardised Tests of Musical Intelligence [68], Measures of Musical Abilities [69] and Musical Aptitude Profile [70], while later-developed musical aptitude tests are the Advanced Measures of Musical Audiation [65] or more recently developed test batteries such as the Profile of Music Perception Skills [71]. Most musical ability measures are perception tests and include rhythmical and tonal dimensions. The AMMA test has been used widely in the scientific field to assess musical ability. The test consists of paired musical statements where either rhythmic, tonal or no changes can occur. The statements are short melodies and considered to be more complex compared to other musical ability measures and therefore it has also been criticized. Thus, it has been suggested that complex and difficult musical stimuli may measure a combination of skills [71]. It could be suggested that complex musical stimuli processing is a rather attention-demanding task. Indeed, music-educated people have been shown to have superior attentional skills [72]. In this investigation, we wanted to use the AMMA test for several reasons since we used it in previous investigations for comparability and, secondly, participants diagnosed with ADHD showed only in the higher-order auditory processing deficits.

While there are multiple musical ability measures available, a considerably lower number of language ability tests that focus on initial learning processes exist and are available. As already mentioned, the LLAMA test focuses on a British Columbian Indian language and has four different subsections which focus on vocabulary learning, sound recognition, sound–symbol correspondence and grammar [62]. Language ability measures such as the Pimsleur Language Aptitude Battery [73] also include sound discrimination tasks in Ewe, and the VORD test uses Turkish as language stimuli since Turkish is typically different to other Western European languages [74]. However, except for the LLAMA test most of the measures are either not available or only consist of a single language which is why we used an adapted version of our recently developed speech perception measurement [22,31,75] which was developed to be a speech perception measure pendant to the AMMA test [31,75]. It has a similar structure to the AMMA test and consists of lengthy strings followed by one or more responses which were either included in the strings participants heard before or not and additionally comprises simple and difficult tasks that increase in complexity.

The phonological STM is another capacity which has been associated with musical [22,30,33] and language ability [76,77,78,79,80,81]. Language learners with enhanced STM capacity are also those who remember, retrieve and incorporate utterances faster [82]. Therefore, the phonological STM capacity is one of the most important predictors explaining individual differences in initial foreign-language-learning settings [83,84]. At the other extreme, impairment status has provided evidence that individuals showing deficits of the phonological STM capacity also show poor foreign language performance [85,86]. Studies on children diagnosed with ADHD provided evidence for deficits of the phonological short term [18,19], while for adults with ADHD less clear results were presented [13]. It could be suggested that STM capacity of individuals diagnosed with ADHD improves with age and probably does not persist into adulthood, contrary to working memory capacity. This may be related to the nature of tasks as, while phonological STM capacity is usually assessed by using series of digits or non-words which have to be recalled and remembered and can be considered to be simple tasks [87,88,89,90,91], working memory capacity measures are of a dual nature, which entails that information has to be both processed and stored [87]. Measures of STM capacity should be included in language and music research since STM capacity deficits may also be responsible for poorer music and language performance and are therefore included in this investigation.

In previous studies, we noted that children with ADHD predominantly show deficits in higher-order processing such as the ability to perceive rhythmic and tonal changes in complex melodies [27]. In a follow-up study, we also noted that musical performance (rhythmic and pitch improvisation) of subjects diagnosed with ADHD was considerably lower than that of the controls [9]. In this study, we wanted to assess individual differences in the perception of music in the same way we did in previous research. In addition, we integrated two further measures that were found to be interrelated to musical abilities, namely, speech perception and STM capacity measures. We recruited young adults with ADHD, as well as two control groups. One was educated in music (music-educated participants) and the second was not educated in music (musically naïve participants). We wanted to address the following research questions. First, we wanted to know whether young adults diagnosed with ADHD also show deficits in the ability to discriminate complex melodies compared to controls and to find out whether previous findings could be replicated when we test young adults. As found in previous research on children and adolescents [26,27], we suggested that young adults with ADHD perform worse in complex musicality measures (Q1). Second, we wanted to assess whether individuals with ADHD also show lower performance in STM ability and the ability to discriminate unfamiliar foreign language material in simple and difficult conditions. We suggested that individuals with ADHD perform worse in the language tasks compared to the control groups since previous research on healthy adults [33,75,92] has shown that language ability and also enhanced STM capacity are intercorrelated with elaborate musical skills (Q2). Third, we also wanted to assess whether the self-estimation variables of the language and music performances differed across groups since we expected individuals diagnosed with ADHD to overestimate their performance as often seen in previous studies [21,24]. Therefore, we also included a music-educated control group in the study since previous studies have shown that individuals with musical training assess their performances very well [22,23,75]. We suggested that if the self-estimation scores of the participants with ADHD show remarkable differences from both control groups relative to the performance scores, it can be suggested that young adults with ADHD misinterpret or overestimate their own performance (Q3).

## 2. Materials and Methods

### 2.1. Participants

We recruited 75 young adults who voluntarily participated in this study. We established certain music and language criteria for the participants. We had three specific music criteria for the selection of the participants since we wanted to divide participants with low musical education and practice (musically naïve participants *n* = 25), those with high musical practice and education (music-educated participants *n* = 25) and individuals diagnosed with ADHD (*n* = 25) with low musical practice and education.

The description for musically naïve participants outlined that they should not be capable of playing a musical instrument or received music lessons on a regular basis beside the general music education which individuals receive in the school environment such as learning the recorder. The same criteria were chosen for the ADHD group. The description of the music-educated participants included that they had received formal music education beside the school environment for at least five years in a row before testing took place. In addition, we also had specific language criteria for the participation in this study. They were the following: All participants should have monolingually grown up as German native speakers who did not receive lessons in or have had knowledge of Thai, Tagalog, Mandarin, Farsi and/or Japanese. As research has shown that the number of foreign languages spoken can also influence individual differences in foreign language capacity for other languages [22,53], we selected participants who did not speak more than three foreign languages (English as the first foreign language, and second one or two other foreign language such as French, Italian or Spanish which was only spoken at the A1 or A2 level).

The mean age of the ADHD group was M = 20.04, SD = 1.62, that of the musically naïve participants M = 20.04, SD = 2.26 and that of the music-educated participants M = 21.00, SD = 0.96. We also collected the highest completed level of education status of the participants and the highest completed level of education status of their parents. In the ADHD group, 5 participants possessed A-levels while 20 had as the highest level of education the main general secondary school certificate. In the musically naïve group, 8 participants possessed A-levels, whereas 17 had as the highest level of education the main general secondary school certificate. In the music-educated group, 7 participants possessed A-levels, while 18 had as the highest level of education the main general secondary school certificate. As the participants were young adults, we suggested that the educational status of the participants may be less informative to provide accurate information about their socio-economic status. Therefore, we used the parental educational background to assess the socioeconomic status (SES) of the participants. Following the classification of UNESCO, the border between low and high SES is assigned at the ISCED-97 level 3a which represents A-levels and/or equivalents. This means participants whose parents had an educational status below level 3a were considered to has a low SES, while participants at level 3a and above were considered as having high SES [93]. We decided to define that if one of the parents had an educational level above 3a the participants would be considered as having high SES, while if both parents had a completed highest level of education below level 3a, the participants would be considered as having low SES. In the ADHD group, 7 participants were categorized as low SES, while 18 were categorized as high SES. In the musically naïve group, 10 participants were categorized as low SES, while 15 were categorized as high SES. In the music-educated group, 8 participants were categorized as low SES, while 17 were categorized as high SES.

In order to outline whether there is an association between the three groups (ADHD, musically naïve and music-educated groups) and the SES status, we performed a chi-square test. Chi-square analysis has revealed that there was no association between SES and the categorical variable (ADHD, musically naïve and music-educated groups). χ^2^(2) = 0.84 *p* = 0.75.

Although gender differences play a role in multiple domains, we did not expect gender to be a crucial parameter in this study. In previous research with a sample of more than 400 participants, gender differences were not detected for ability measures that focus on unfamiliar language and music tasks (e.g., unfamiliar language perception, or unfamiliar song-singing tasks) [22]. It was concluded that as long as tasks are not influenced by educational differences, gender does not have an impact on performances.

In this study, in the ADHD group, 8 participants were female and 17 participants were male. In the musically naïve group, 12 participants were female and 13 participants were male. In the music-educated group, 14 participants were female and 11 participants were male. In order to outline whether there was an association between the three groups (ADHD, musically naïve and music-educated groups) and gender, we performed a chi-square test. Chi-square analysis has revealed that there was no association between SES and the categorical variable (ADHD, musically naïve and music-educated groups). χ^2^(2) = 3.01 *p* = 0.27.

The participants had no hearing and cognitive impairments. Informed consent was obtained from all subjects involved in the study. This study was approved by a medical ethics committee (2-PĒK-4/3/2022). Results indicated that none of the main variables were influenced by gender differences.

### 2.2. Speech Perception

To assess individual differences in phonetic perception ability, we used a speech perception measure which assesses individual differences in the ability to discriminate unfamiliar languages (Thai, Tagalog, Mandarin, Farsi, Japanese). We used an adapted version of previous investigations [22,31,75]. The language measurement consists of two difficulty levels (simple and difficult). The two levels of the speech perception measures have the same fundamental basis. In the testing condition, the participants are introduced to a sequence of speech strings (Stimlines) which consist of eight different words or short phrases (Stims) followed by a response (Stimcompare) which was either included in the string the participants had listened to before or not.

The Stims are separated by a pause of 50 ms and the Stimcompare is separated from the string by a longer pause (2 s) and is additionally indicated by a color change on the screen. After the participants had listened to a sample, they had to indicate whether the Stimcompare was contained in the Stim they had just heard or not. If the Stimcompare was part of the Stimline, the participants had to click the “correct” button. If the Stimcompare was not part of the Stimline, the participants had to click the “incorrect” button. In addition, the participants were asked to provide answers about the degree of certainty of their answers which is the self-estimation language variable in this study. They could choose from three options, “very sure”, “rather sure” and “can’t tell”.

The two difficulty levels of the test include a Stimcompare consisting of different numbers of Stims. In the simple condition, the Stimcompare consists of 1 Stim, while in the difficult condition the Stimcompare consists of either 2 or 3 Stims. The difficult condition meant that the Stimcompare was only correct if all Stims were included in the Stimline they had listened to before.

The simple condition consists of 10 items, while the difficult condition consists of 16 items. This measure generates four main scores. First, two separate scores of the simple and the difficult condition, a composite score which consists of both conditions (simple and difficult) and a composite score of the degree of certainty.

This speech perception task provides information about individual differences in the ability to discriminate unfamiliar languages. This measure simulates early language learning settings when the language input is poor in semantic content [31,61,75] and represents an equivalent to musical aptitude measures which are based on the assumption that the stimuli (e.g., melody) listened to have not been heard before. Similarly, this language measure reduces the influence of foreign language education since the stimuli are completely unfamiliar to the participants.

### 2.3. Short-Term Memory (STM) Test

Participants’ phonological STM was assessed by using forward and backward digit span tasks [94]. Participants were asked to reproduce an increasing sequence of numbers which were presented auditorily in either a forward or backward order. First, they had to perform the forward span and afterwards the backward span. The task was computerized and executed online. The forward span version included three to nine digits, whereas the backward span version included two to eight digits. For each correct answer, the participants received one point and therefore the STM score represents the number of correctly answered items of both the forward and the backward span. The STM test was included for several reasons. First, music and language variables are related to STM capacity and, secondly, we also wanted to ensure that individual differences in music and language tasks are not influenced by a much lower STM capacity in the ADHD group.

### 2.4. Music Perception Ability

To assess participants’ music perception abilities, the Advanced Measures of Music Audiation (AMMA, Gordon, 1989) test was administered. This musical ability measure included participants determining whether paired musical statements are identical or contain a rhythmic or tonal difference in the second musical statement. Rhythmic changes can be represented as tempo, meter or duration, while tonal differences are represented as different notes. The test consists of 33 items. Participants started with three practice trials before proceeding to the remaining 30 trials. These 30 trials consist of 10 identical pairs of statements, 10 pairs with rhythmic differences and 10 pairs with tonal differences. In addition, the participants were also asked to assess how well they performed in the music perception test by choosing from five options, “very poor”, “poor”, “average”, “good”, and “very good” which is the “self-estimation musical ability” score.

### 2.5. Procedure

The testing of the participants included several steps. First, participants were asked to provide background information which was crucial for this study. This was carried out online before the main measures were tested. This approach was also used to select appropriate participants for this study. Participants who were suitable were then additionally invited to participate in the main study in a laboratory. There, the three main measures (speech perception, AMMA and STM tasks) were performed. The tasks were all computerized and administered online to ensure that the conditions were as equal as possible. The participants completed the familiarization of the tasks together with the experimenter to ensure that all participants understood their tasks precisely.

The speech perception task lasted around 15 min, the AMMA test 20 min and the STM measure on average 12 min. Participants had a 10 min break between each of the tests. The ordering of the tests was the musicality measures (AMMA) first, followed by the speech perception measure and finally the STM capacity test.

### 2.6. Statistical Analysis

The statistical analysis consists of four main sections. First, we calculated descriptive statistics for the three groups separately (see Table 1). Next, we used the total scores of the music and language self-estimation variables (degree of certainty and self-estimation musical ability) and the perceptual musicality, language and the STM measurements and performed a MANOVA, followed by a discriminant analysis. This aimed at showing which of our variables split our groups up best. In previous research, we demonstrated that individuals with ADHD showed lower musical ability in complex music processing compared to musically naïve participants [27], therefore, we wanted to focus on the two language conditions in more detail and also performed separate ANOVAs for the simple and the complex language tasks. In addition, we also performed separate ANOVAs to assess STM capacity in more detail.

Although we did not work on correlational analyses between the self-estimation variables and the respective performances, we provide two correlation tables of the self-estimation variables and the performances in the supplement for transparency reasons. We run the analysis one time with all participants, and a second time for the healthy subjects only, in order to outline that the relationship between the self-estimation variables and their respective performances improves when the ADHD group has been removed (see Appendix A).

## 3. Results

### 3.1. Descriptive Statistics

Table 1 illustrates the means and standard deviations of the total scores of the perceptual language and musical variables, the self-estimation variables and STM capacity.

### 3.2. MANOVA and Discriminant Analysis

We performed a MANOVA to reveal whether our language and music perception variables differ in their mean values based on the grouping variable (ADHD, musically naïve participants and music-educated participants). Using Pillai’s trace, there was a significant effect of participants’ self-estimation on their language and music performances and their language and music perception ability, V = 0.81, *F*(10, 1389) = 9.23 *p* < 0.001. Since the MANOVA was significant, we ran a discriminant analysis as a follow-up analysis.

Discriminant analysis revealed two discriminant functions. The first explained 90.4% of the variance, canonical R² = 0.64, whereas the second explained only 9.6%, canonical R² = 0.16. In combination, these discriminant functions significantly discriminated the groups, Λ = 0.29, χ^2^(10) = 84.45, *p* < 0.001. When removing the first function, the second function also significantly differentiates the three groups Λ = 0.84, χ^2^(4) = 12,25, *p* = 0.016.

We chose a statistically recommended cutoff of 0.4 to decide which of the standardized discriminant coefficients were large enough to be significant [95]. The correlations between the outcomes and the discriminant functions show that the loads on the first function are rather high for the musicality ability measure (r = 0.67), followed by the STM measure (r = 0.43) and the language perception measure (r = 0.40). However, the musical ability measure, the STM measure and the language perception measure show considerably lower loads on the second function (see Appendix A).

In marked contrast to the first discriminant function, the loads on the second discriminant function are rather high for the self-estimation variables self-estimation musical ability (r = 0.79) and the degree of certainty of the language performances (r = −0.56), while both showed low loads on the first function (see Appendix A).

The first function shows that the variables perceptual musical ability, STM capacity and perceptual language ability separate the ADHD group from the musically naïve and music-educated participants best. Looking at the means of the three groups, the music-educated participants have the highest mean values in the perceptual musical ability score (M = 0.64, SD = 0.09), the STM score (M = 15.55, SD = 2.22) and the perceptual language ability score (M = 61.52, SD = 7.91), followed by the musically naïve group (perceptual language ability score M = 0.56, SD = 0.09; the STM score M = 13.88, SD = 1.88; perceptual musical ability score: M = 52.16, SD = 5.47) and the ADHD group (perceptual musical ability score M = 46.72, SD = 6.72; the STM score M = 12.76, SD = 1.92; perceptual language ability score M = 0.52, SD = 0.09).

Taking the two musical and language self-estimation variables into account, the second function shows that the musically naïve participants can be separated from both other groups, the ADHD and the music-educated group, showing that the ADHD group and the music-educated participants estimate their musical ability and language performance similarly well in contrast to the musically naïve participants. The ADHD group have the highest means for the musical self-estimation variable (M = 3.24, SD = 1.05) and the degree of certainty (M = 2,16 SD = 0.26), followed by the music-educated participants (musical self-estimation M = 3.04, SD = 1.02 and the degree of certainty M = 1.93, SD = 0.39) and the musically naïve participants (musical self-estimation M = 2.44, SD = 0.77 and the degree of certainty M = 1.8, SD = 0.49). Findings indicate that even though the language, STM and music performance of the ADHD group is lower than of the musically naïve participants and music-educated participants, they estimate their musical ability and language ability similarly to the music-educated participants.

The discriminant plot illustrates the finding (see Figure 1).

### 3.3. Separate ANOVAs: Language Perception Ability

Furthermore, we also run two separate ANOVAs for the STM capacity and simple and complex language perception measurement to outline whether differences across groups in these conditions were detected.

A separate ANOVA for the simple speech perception measure revealed that although the mean values of the ADHD group (M = 0.62, SD = 0.11) and the musically naïve (M = 0.62, SD = 0.15) differed from that of the music-educated participants (M = 0.67, SD = 0.15), the difference was statistically non-significant *F*(2, 72) = 1.00, *p* > 0.37. However, there was a significant effect on the difficult speech perception task and group membership, F(2, 72) = 21.64, *p* < 0.001, ω = 0.59. There was also a linear trend, *F*(1, 72) = 43,15, *p* < 0.001, ω = 0.60, indicating that music-educated participants performed best followed by the musically naïve participants and individuals diagnosed with ADHD. Planned contrasts revealed that the music-educated participants performed significantly better in the complex language task than the musically naïve participants, *t*(72) = −3.58, *p* < 0.001, r = 0.39, and the ADHD group, *t*(72) = −6.57, *p* < 0.001, r = 0.61. Planned contrasts also revealed that the musically naïve participants performed significantly better in the complex language task than the ADHD group, *t*(72) = 2.99, *p* < 0.004, r = 0.33. The linear trend shows that the music-educated participants (M = 0.74, SD = 0.08) performed best, followed by the musically naïve participants (M = 0.65, SD = 0.08) and individuals with ADHD (M = 0.58, SD = 0.10).

We also detected a significant effect on STM capacity and group membership, *F*(2, 72) = 12.27, *p* < 0.001, ω = 0.32. There was also a linear trend, *F*(1, 72) = 24.22, *p* < 0.001, ω = 0.48, indicating that music-educated participants performed best followed by the musically naïve participants and individuals diagnosed with ADHD. However, planned contrasts revealed that there was only a statistically significant difference between the music-educated participants and the musically naïve participants, *t*(72) = 2.95, *p* < 0.012, r = 0.32, and the ADHD group, *t*(72) = −4.9, *p* < 0.001, r = 0.50, while no significant difference between the musically naïve and the music-educated participants was detected.

## 4. Discussion

In this investigation, we addressed three research questions. Based on previous investigations, we knew that children and adolescents diagnosed with ADHD show deficits in the ability to discriminate complex melodies compared to controls. In this study, we wanted to replicate the research design with young adults (Q1). Results of the discriminant analysis (see Figure 1 and Appendix A) showed that the ability to discriminate deviations in complex melodies distinguished the young adult ADHD group from both controls. This shows the ability to discriminate complex melodies is lower in individuals with ADHD and this seems to persist into young adulthood.

In addition, we wanted to test whether individuals with ADHD perform lower in STM capacity and the ability to discriminate unfamiliar language material in simple and difficult conditions (Q2) and we also wanted to provide information on music and language self-estimation parameters of the participants (Q3). Discriminant analysis revealed that the ability to discriminate speech also differentiated both control groups from the ADHD group. This finding was expected. However, when the language perception measures were divided into simple and difficult tasks to increase complexity, results indicated that the ADHD group did not perform significantly worse when the language tasks were simple, while the opposite was found when the language tasks were difficult. The results of the self-assessment parameters confirm the findings of previous studies that people diagnosed with ADHD cannot accurately assess their own performances. Based on the results of our investigation, we will discuss the three research questions separately.

### 4.1. Musicality and ADHD

The findings of our study show that young adults with ADHD also perform worse in complex musicality tasks than the musically naïve and the music-educated participants (Q1). Previous research has shown that children show lower ability in complex musical stimuli processing [27]. Adolescents diagnosed with ADHD also indicated inferior rhythmical and pitch improvisation performance in music compared to their non-affected peers [26]. Most importantly, it has been suggested that this may stem from auditory deficits which affect perception skills of temporal auditory information such as duration discrimination, timing skills and the ability to perceive rhythmic information [43,44]. Auditory deficits could also be a probable reason why young adults diagnosed with ADHD perform worse in complex musical tasks—meaning this phenomenon probably persists into young adulthood. Another explanation for lower performance in complex music stimuli discrimination of young adults with ADHD could be based on attentional deficits. In previous research, we also noted that differences between subjects with and without ADHD were not reported for simple music-related tasks such as detecting high versus low and short versus long tones [27]. Criticism on the AMMA test suggested that complex melodies, as used for this test, measure a combination of skills and therefore are not pure musicality measures. For instance, the assessment of rhythm is based on paired musical statements that also consist of complex melodic information [71] and therefore requires higher loads of attentional control. It is suggested that musical training improves attentional skills [72,96]. For instance, musical training studies have shown that the performance of attention-demanding tasks is indeed enhanced in individuals with extensive musical training [97,98,99,100]. In our study, we found a linear trend that indicated that the music-educated participants performed best in the musicality test, followed by the musically naïve participants and the ADHD group. Thus, it can be suggested that the findings of the musicality measure could partly be a result of the degree of attentional control.

### 4.2. Speech Perception, the Phonological STM and ADHD

In this study, we also assessed whether the ability to discriminate unfamiliar speech is different for the ADHD group and both controls (Q2). In general, discriminant analysis has shown that foreign language perception ability and STM capacity separate the ADHD group from the two control groups. However, when we looked at both measures in more detail, the performance differences between the groups were similar to what we observed in this and in previous research on musicality and ADHD [27]. When the stimulus is complex and difficult, individuals with ADHD show inferior performance compared to controls. We suggested that if individuals with ADHD perform differently in simple and complex conditions, we can suggest having found a perceptual issue. Indeed, we found this when we analyzed both language conditions separately. The separate ANOVAs of the simple and difficult language perception tasks showed that group differences between the two control groups and the ADHD group were only observed when the language tasks were more difficult, while in the simple condition no statistically significant difference across all three groups was detected (see Section 3.3).

A separate ANOVA on STM capacity provided that a significant difference was only observed between the music-educated group, the musically naïve participants and the ADHD group, while none was found between the ADHD group and musically naïve participants. When closely looking, the findings of the STM capacity and the language perception tasks fit well into observations of current research for several reasons.

The phonological STM capacity was assessed by using a digit span and, compared to working memory tasks, digit spans are considered to be simple [87,88,89,90,91], while working memory measures are complex since they are of a dual nature, which entails that information has to be both processed and stored [87]. Studies on adults diagnosed with ADHD that assessed the phonological STM reported small to moderate deficits compared to controls [13], while for working memory there is a common agreement that working memory capacity deficits of individuals with ADHD persist into adulthood [7,11,13,14,15,16,17]. The difference between simple and difficult tasks of the language perception measure shows similarities to the differentiation of STM and working memory tests. Our language perception measure consists of strings of language fragments followed by a response. In the simple condition, the response consists of a single utterance, while in the difficult condition the response consists of either two or three utterances. In the difficult condition, listeners need to cope with additional scenarios compared to the simple condition. They need to decide whether all of the utterances were included in the string they listened to before or not. This increases the difficulty and complexity level of the task because responses could also include one or two incorrect responses which were additionally mixed up with correct responses.

Therefore, the difficult language perception task requires more cognitive capacity and attentional control and is comparable in difficulty and complexity to working memory tasks. This finding could be taken as an indicator that the discrimination of auditory stimuli is not impaired in young adults diagnosed with ADHD, as long as the stimuli are simple, whereas the opposite is the case for difficult and complex stimuli, regardless of whether the stimuli are of musical or language origin.

One, however, could suggest that our argumentation would infer that STM capacity is not, or just minimally, different for both control groups, which was not found. However, the superior performance of the music-educated group compared to individuals with ADHD may be due to a positive transfer from musical training to STM capacity. In previous research, we noted that STM capacity was associated with musical status [33], musical ability [33,101] and the singing of familiar [22,30,33] and unfamiliar songs [30]. The findings of our study suggest that perceptual music and language deficits persist into adulthood if the stimuli are difficult or complex but not if the stimuli are simple. This also has implications on music and language pedagogy. Hence, the music and foreign language input should be simple and less complex to facilitate learning processes of individuals with ADHD.

### 4.3. Self-Estimation and ADHD

We also included two self-estimation parameters in the research design in which the participants had to verify the degree of certainty of their language performance and their self-estimation of their musical ability (Q3). Discriminant analysis showed that both self-estimation variables separated the music-educated participants and the ADHD group from the non-musicians. Although the ADHD group performed lower than the music-educated and the musically naïve participants in both the language and music tasks, the ADHD group reported that their music and language performances were as good as those of the music-educated group. Research has shown that self-assessment of musical ability is a meaningful part of musicianship [102]. Previous research has shown that the musical performance of musicians and healthy non-musicians is remarkably correlated to their own self-evaluation [22,102]. To assess whether our self-estimation variables were meaningful, we also provide correlations in the supplement which show that self-estimation parameters improve remarkably when the ADHD group has been removed (see Appendix A). This discrepancy between self-perceptions and performance of individuals with ADHD has also been referred to as the positive illusory bias [21,24]. Studies suggest that adults diagnosed with ADHD have significantly lower levels of self-esteem and self-efficacy than comparable healthy adults [103,104]. The overrating of performances has been suggested to operate as a defense mechanism to show confidence [105] and to protect the self-image [25]. Research has even put forward that individuals with ADHD tend to overestimate their competence most in the areas they show the greatest deficits [106,107]. Following the current literature, the discrepancy between self-assessment and performance in individuals diagnosed with ADHD appears to be a creative conscious compensatory strategy rather than an inability to recognize their own deficits. In addition, learning progress is associated with being able to evaluate one’s own performance in order to reach a goal [108]. However, studies have also shown that people with ADHD provide more realistic self-assessment when their self-image is strengthened [109]. Therefore, individuals with ADHD require individualized plans that do not only facilitate processes to improve performances in music and language but also metacognitive training that involves self-assessment and ways to strengthen the self-image.

This study also has limitations. The main aim of the study was to look at music and language perception ability. This means that this study only provides limited insights into deficits of foreign language capacity and musical skills of individuals with ADHD. Future studies have to consider larger sample sizes and include different tasks which assess various domains in language and music to understand the impact of ADHD more precisely. In addition, future studies should also address whether the overrating of performances in individuals with ADHD also encompasses tasks for which they do not show any deficits.

## 5. Conclusions

In this study, we assess individual differences in STM capacity, the ability to discriminate music and language stimuli and the ability to self-estimate the performance. Our findings suggest that young adults diagnosed with ADHD show deficits in music and unfamiliar language perception when the stimuli and tasks are complex and difficult but not if the stimuli and tasks are simple. We also suggest that the STM capacity of young adults diagnosed with ADHD does not show impairments and STM capacity deficits do not persist into young adulthood. In addition, we also introduced self-estimation criteria of the language and music measures and noted that even though the ADHD group showed inferior performance in the language and music tasks, they estimated their performance as being as good as that of the best performers (music-educated group).

As a result, specialized music and language learning programs for individuals diagnosed with ADHD should consider different domains. Based on our findings, teaching methods for individuals with ADHD should focus on how difficult and complex music and language tasks could be presented in a simple way. In addition, metacognitive training should be included that involves aspects to strengthen self-esteem and self-image and facilitate the development of realistic self-assessment. In this respect, a musical training program would be highly beneficial for many reasons. First, musical training improves perceptual skills in both domains, music and language. Second, in music education self-assessment of one’s own musical abilities is an important part of musical training. Third, musical training increases self-confidence and self-esteem [110].

## Figures and Tables

**Figure 1 ijerph-20-06841-f001:**
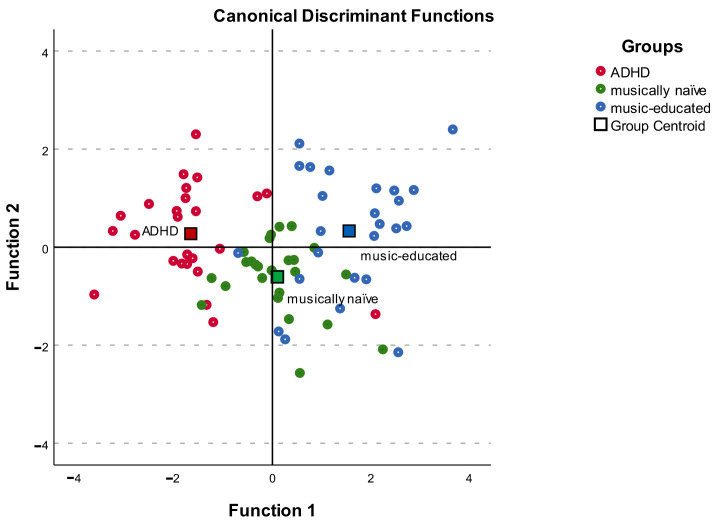
The discriminant plot illustrates the discriminant function of the variables of the investigation. Function 1 discriminates the ADHD group from the musically naïve and the music-educated groups and function 2 discriminates the musically naïve participants from both other groups (ADHD and music-educated group).

**Table 1 ijerph-20-06841-t001:** Descriptive statistics of the variables of this investigation.

Variable	Group	Mean (M)	Standard Deviation (SD)
Perceptual language ability			
	ADHD	0.52	0.09
	Musically naïve	0.56	0.09
	Music-educated	0.64	0.09
Perceptual musical ability			
	ADHD	46.72	6.72
	Musically naïve	52.16	5.47
	Music-educated	61.52	7.91
Self-estimation musical ability			
	ADHD	3.24	1.05
	Musically naïve	2.44	0.77
	Music-educated	3.04	1.02
Degree of certainty			
	ADHD	2.16	0.26
	Musically naïve	1.80	0.49
	Music-educated	1.93	0.40
Short-term memory capacity			
	ADHD	12.76	1.92
	Musically naïve	13.88	1.88
	Music-educated	15.56	2.22

## Data Availability

Data are contained in the article or Appendix A.

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
