# Peer review of "Misjudgement of One’s Own Performance? Exploring Attention Deficit (Hyperactivity) Disorder (ADHD) and Individual Difference in Complex Music and Foreign Language Perception"

_ijerph, 2023, doi:10.3390/ijerph20196841_

Round 1

Reviewer 1 Report

This contribute offers an interesting overview of the cognitive processes shared by language and music learning, with a specific focus on the role of attention and memory, in typical and atypical populations.

The introduction is well organized, and the main references of the literature in the field were considered. There are some minor points to be amended in the Materials and Methods section:

1.       P.4: The three groups of participants should be described in details according to (at least) gender distribution, mean age, education level, and socio-economic-status. It is important to be sure that ADHD actually is the main variable affecting results, and that other potentially relevant variables are matched between groups;

2.        P.6: After presenting measures, the procedure should be described: how the sample was recruited, in which order the tasks were presented, how much time was spent for the whole assessment of each subject? The number of the title “Statistical analysis” should be revised.

 Even though the whole research addresses interesting questions, the conclusions are not completely consistent with the results. In particular, it is too reductive to consider the ADHD people's overestimation of language and music learning ability as a good reason for avoiding encouragement and support to them in learning processes. On the opposite, it might be considered a suggestion to develop metacognitive trainings for ADHD children and adults and to support their self esteem, by promoting the development of strategic behaviors since childhood.

Author Response

Reviewer 1

This contribute offers an interesting overview of the cognitive processes shared by language and music learning, with a specific focus on the role of attention and memory, in typical and atypical populations.

The introduction is well organized, and the main references of the literature in the field were considered. There are some minor points to be amended in the Materials and Methods section:

Dear reviewer many thanks for your kind and valuable comments on our manuscript. We are very thankful for them and tried to integrate your kind suggestions into our manuscript and hope to have addressed your suggestions satisfactorily. We answered all your questions point by point and we also added the new passages here as well as we included the line number of the manuscript where we marked larger changed parts in grey. We hope that this saves you time.

  1. 4: The three groups of participants should be described in details according to (at least) gender distribution, mean age, education level, and socio-economic-status. It is important to be sure that ADHD actually is the main variable affecting results, and that other potentially relevant variables are matched between groups;

Many thanks for this comment. We included the information which you suggested and want to thank you for your input. We included the mean age of each of the three groups. In addition, we included the educational status of the participants and used the educational status of the parents to divide them into high and low SES. We also performed a chi-square test to determine whether there was an association between the categorical variable (ADHD, music-educated and musically naïve participants) and the SES status. For gender and the three groups we also performed a chi-square test. However, we did not expect gender differences at all. In previous research, we collected large samples with the same language measure as used in this investigation and looked at gender from multiple perspectives. Interestingly, we did never detect gender differences for the language perception tasks which were completely unfamiliar to the participants (e.g unfamiliar speech perception tasks etc.). This suggests that gender has no influence on the potential to learn foreign languages. We cited previous research which consisted of 400 participants (children, adolescents and adults) in the text (see lines 264-269 and below).

“Although gender differences play a role in multiple domains, we did not expect that gender is a crucial parameter in this study. In previous research with a sample of more than 400 participants, gender differences were not detected for ability measures that focus on unfamiliar language and music tasks (e.g. unfamiliar language perception, or unfamiliar song singing tasks) (22). It was concluded that as long as tasks are less influenced by educational differences, gender does not have an impact on performances.”

1a) In addition, we also run t-tests for gender and SES and provided on the performance variables for each group and produced tables. The results show that SES and gender has no influence on the performances. We included this only for you but not in the manuscript.

For gender see table 1, the descriptives and 2-4 the t-test at the end of this document on page 7 ff.

For SES see table 5, the descriptives and 6-8 the t-tests at the end of this document on page 9 ff.

1b) Since we provided chi-square tests in the main text, we found that there was no need to keep the gender differences of all participants (table S1) in the supplement and deleted this information as it would have confused the readers why there is a second gender analysis. We hope this is fine for you.

1c) The relevant information which was added in the text follows below. We hope that the changes we made are satisfactorily.

See also lines 235-276 in the manuscript and below

“The mean age of the ADHD group was M = 20.04, SD = 1.62, of the musically naïve participants M = 20.04, SD = 2.26, and of the music-educated participants M = 21.00, SD = 0,96. We also collected the highest completed level of educational status of the participants and the highest completed level of the educational status of their parents. In the ADHD group 5 participants possess the A-levels while 20 had as the highest level of education the main general secondary school certificate. In the musically naïve group 8 participants possess the A-levels, whereas 17 had as the highest level of education the main general secondary school certificate. In the music-educated group 7 participants possess the A-levels, while 18 had as the highest level of education the main general secondary school certificate. As the participants were young adults, we suggested that the educational status of the participants may be less informative to provide accurate information about their socio-economic status. Therefore, we used the parental educational background to assess the socioeconomic status (SES) of the participants. Following the classification of the UNESCO, the border between low and high SES is assigned at the ISCED-97 level 3a which represents the A-levels and/or equivalents. This means participants whose parents had an educational status below the level 3a were considered to belong to low SES, while participants at level 3a and above were considered as high SES (92). We decided to define that if one of the parents has an educational level above 3a the participants will be considered as high SES, while if both parents had a completed highest level of education below the level 3a, the participants will be considered as low SES.  In the ADHD group 7 participants were categorized as low SES, while 18 were categorized as high SES. In the musically naïve group 10 participants were categorized as low SES, while 15 were categorized as high SES. In the music-educated group 8 participants were categorized as low SES, while 17 were categorized as high SES.

In order to outline whether there is an association between the three groups (ADHD, musically naïve and the music-educated) and the SES status, we performed a chi-square tests. Chi-square analysis has revealed that there was no association between SES and the categorical variable (ADHD, musically naïve and the music-educated). χ²(2) = 0.84 p = 0.75.

Although gender differences play a role in multiple domains, we did not expect that gender is a crucial parameter in this study. In previous research with a sample of more than 400 participants, gender differences were not detected for ability measures that focus on unfamiliar language and music tasks (e.g. unfamiliar language perception, or unfamiliar song singing tasks) (22). It was concluded that as long as tasks are less influenced by educational differences, gender does not have an impact on performances.

In this study, in the ADHD group, 8 participants were female and 17 participants were male. In the musically naïve group, 12 participants were female and 13 participants were male. In the music-educated group, 14 participants were female and 11 participants were male. In order to outline whether there was an association between the three groups (ADHD, musically naïve and the music-educated) and gender, we performed a chi-square tests. Chi-square analysis has revealed that there was no association between SES and the categorical variable (ADHD, musically naïve and the music-educated). χ²(2) = 3.01 p = 0.27.”

  1. 6: After presenting measures, the procedure should be described: how the sample was recruited, in which order the tasks were presented, how much time was spent for the whole assessment of each subject?

Many thanks for this comment. We included a passage that describes the procedure of participants testing.

see lines 350-364 in the manuscript and below:

“2.5 Procedure

The testing of the participants included several steps. First, participants were introduced to provide background information which were crucial for this study. This was done online before the main measures were tested. This approach was also decided to select appropriate participants for this study. Participants who were suitable were then additionally invited to participate in the main study in a laboratory. There the three main measures (speech perception, AMMA and STM tasks) were performed. The tasks were all computerized and administered online to ensure that the conditions were as equal as possible. The participants completed the familiarization of the tasks together with the experimenter to ensure that all participants understood their tasks precisely.

The speech perception task lasted around 15 minutes, the AMMA test 20 minutes and the STM measure on average 12 minutes. Participants had a 10-minute break between each of the tests. The ordering of the tests was that the musicality measures (AMMA) were performed first, followed by the speech perception measure and finally the STM capacity test.”

  1. The number of the title “Statistical analysis” should be revised.

Thank you we corrected it from 2.4 to 2.6.

  1. Even though the whole research addresses interesting questions, the conclusions are not completely consistent with the results. In particular, it is too reductive to consider the ADHD people's overestimation of language and music learning ability as a good reason for avoiding encouragement and support to them in learning processes.

Many thanks for this comment. This is indeed right and was not well formulated and we deleted this sentence “…but also reduce settings in which overestimating as a compensatory strategy is beneficial for individuals diagnosed with ADHD.”  We reformulated this short passage and hope this is fine for you.

see lines 608-611 in the manuscript and below:

“In addition, learning progress is associated with being able to evaluate the own performance in order to reach a goal (107). However, studies have also shown that people with ADHD provide more realistic self-assessment when their self-image is strengthened (108). Therefore, individuals possessing ADHD require individualized plans that do not only facilitate processes to improve performances in music and language, but also metacognitive training that involves self-assessment and ways to strengthen the self-image.”

  1. On the opposite, it might be considered a suggestion to develop metacognitive trainings for ADHD children and adults and to support their self-esteem, by promoting the development of strategic behaviors since childhood.

This is indeed right, and we want to thank you for this valuable comment. Thanks to your comment we were able to formulate how a training programme for individuals with ADHD could look like should focus on.

see lines 638-648 in the manuscript and below:

“As a result, specialized music and language learning programs for individuals diagnosed with ADHD should consider different domains.  Based on our findings teaching methods for individuals with ADHD should focus on how difficult and complex music and language tasks could be presented in a simple way. In addition, metacognitive training should be included that involves aspects to strengthen the self-esteem and self-image and facilitates the development of realistic self-assessment. In this respect a musical training program would be highly beneficial for many reasons. First, musical training improves perceptual skills in both domains, music and language. Second, in music education self-assessment of the own musical abilities is an important part of musical training. Third, musical training increases the self-confidence and self-esteem (109).”

Additional tables GENDER:

For the variables in which the participants also had to perform we provided independent t-tests to show that we did not observe mean differences for gender and SES.

Table 1 shows the descriptives of the variables of interest and gender of all groups

Variable

Group

Mean (M)

Standard Deviation (SD)

Perceptual musical ability

ADHD female

43.75

7.13

ADHD male

48.12

6.24

Musically naïve female

50.92

3.73

Musically naïve male

53.31

6.64

Music-educated female

60.43

7.28

Music-educated

male

62.91

8.80

Perceptual language ability

ADHD female

0.50

0.10

ADHD male

0.52

0.09

Musically naïve female

0.57

0.09

Musically naïve male

0.55

0.08

Music-educated female

0.66

0.08

Music-educated male

0.61

0.10

Simple condition

ADHD female

0.62

0.10

ADHD male

0.61

0.12

Musically naïve female

0.67

0.17

Musically naïve male

0.58

0.11

Music-educated female

0.70

0.12

Music-educated male

0.64

0.25

Complex condition

ADHD female

0.55

0.11

ADHD male

0.59

0.09

Musically naïve female

0.64

0.09

Musically naïve male

0.66

0.07

Music-educated female

0.75

0.09

Music-educated male

0.73

0.08

Short term memory capacity

ADHD female

13.50

1.69

ADHD male

12.41

1.97

Musically naïve female

13.92

1.51

Musically naïve male

13.85

2.23

Music-educated female

15.64

2.44

Music-educated male

15.46

2.02

Table 2 shows t-tests of the variables of interest of the ADHD group and gender.

Variable

t

df

p

Mean Difference

Perceptual musical ability

Equal variances assumed

1,561

23

0,132

4,36765

Equal variances not assumed

1,486

12,270

0,163

4,36765

Perceptual language ability

Equal variances assumed

0,598

23

0,556

0,02402

Equal variances not assumed

0,576

12,656

0,574

0,02402

STMT

Equal variances assumed

-1,343

23

0,192

-1,08824

Equal variances not assumed

-1,422

15,959

0,174

-1,08824

Simple condition

Equal variances assumed

-0,265

23

0,794

-0,01324

Equal variances not assumed

-0,281

16,112

0,782

-0,01324

Complex condition

Equal variances assumed

0,943

23

0,355

0,04136

Equal variances not assumed

0,908

12,598

0,381

0,04136

Table 3 shows t-tests of the variables of interest of the Musically naïve group and gender.

Variable

t

df

p

Mean Difference

Perceptual musical ability

Equal variances assumed

1,097

23

0,284

2,39103

Equal variances not assumed

1,121

19,162

0,276

2,39103

Perceptual language ability

Equal variances assumed

-0,419

23

0,679

-0,01460

Equal variances not assumed

-0,415

20,811

0,682

-0,01460

STMT

Equal variances assumed

-0,092

23

0,928

-0,07051

Equal variances not assumed

-0,093

21,146

0,927

-0,07051

Simple condition

Equal variances assumed

-1,444

23

0,162

-0,08205

Equal variances not assumed

-1,417

18,099

0,173

-0,08205

Complex condition

Equal variances assumed

0,564

23

0,578

0,01803

Equal variances not assumed

0,559

20,981

0,582

0,01803

Table 4 shows t-tests of the variables of interest of the Music-educated group and gender.

Variable

t

df

Mean Difference

p

Perceptual musical ability

Equal variances assumed

0,772

23

0,448

2,48052

Equal variances not assumed

0,754

19,344

0,460

2,48052

Perceptual language ability

Equal variances assumed

-1,377

23

0,182

-0,05108

Equal variances not assumed

-1,345

19,370

0,194

-0,05108

STMT

Equal variances assumed

-0,206

23

0,838

-0,18831

Equal variances not assumed

-0,211

22,909

0,835

-0,18831

Simple condition

Equal variances assumed

-0,844

23

0,407

-0,06364

Equal variances not assumed

-0,779

13,468

0,450

-0,06364

Complex condition

Equal variances assumed

-0,664

23

0,514

-0,02273

Equal variances not assumed

-0,671

22,443

0,509

-0,02273

Additional tables SES:

Table 5 shows the descriptives of the variables of interest and socioeconomic status (SES) of all groups.

Variable

Group

Mean (M)

Standard Deviation (SD)

Perceptual musical ability

ADHD high SES

47.72

6.54

ADHD low SES

44.14

6.96

Musically naïve high SES

50.93

3.88

Musically naïve low SES

54.00

7.07

Music-educated high SES

62.65

7.73

Music-educated

Low SES

59.13

8.27

Perceptual language ability

ADHD high SES

0.52

0.09

ADHD low SES

0.51

0.12

Musically naïve high SES

0.58

0.08

Musically naïve  low SES

0.53

0.09

Music-educated high SES

0.64

0.10

Music-educated low SES

0.63

0.09

Simple condition

ADHD high SES

0.61

0.12

ADHD low SES

0.63

0.11

Musically naïve high SES

0.67

0.17

Musically naïve low SES

0.64

0.15

Music-educated high SES

0.69

0.20

Music-educated low SES

0.63

0.16

Complex condition

ADHD high SES

0.58

0.09

ADHD low SES

0.55

0.12

Musically naïve high SES

0.66

0.09

Musically naïve  low SES

0.63

0.07

Music-educated high SES

0.73

0.08

Music-educated low SES

0.77

0.09

Short term memory capacity

ADHD high SES

12.50

1.69

ADHD low SES

13.43

2.44

Musically naïve high SES

13.93

1.47

Musically naïve  low SES

13.80

2.44

Music-educated high SES

15.53

2.04

Music-educated low SES

15.63

2.72

Table 6 shows t-tests of the variables of interest of the ADHD group and SES.

Variable

t

df

p

Mean Difference

Perceptual musical ability

Equal variances assumed

1,208

23

0,239

3,57937

Equal variances not assumed

1,174

10,395

0,267

3,57937

Perceptual language ability

Equal variances assumed

0,159

23

0,875

0,00670

Equal variances not assumed

0,139

8,686

0,893

0,00670

STMT

Equal variances assumed

-1,089

23

0,287

-0,92857

Equal variances not assumed

-0,924

8,343

0,381

-0,92857

Simple condition

Equal variances assumed

-0,337

23

0,740

-0,01746

Equal variances not assumed

-0,346

11,637

0,735

-0,01746

Complex condition

Equal variances assumed

0,647

23

0,524

0,02976

Equal variances not assumed

0,598

9,518

0,564

0,02976

Table 7 shows t-tests of the variables of interest of the Musically naïve group and SES.

Variable

t

df

p

Mean Difference

Perceptual musical ability

Equal variances assumed

-1,401

23

0,174

-3,06667

Equal variances not assumed

-1,251

12,651

0,233

-3,06667

Perceptual language ability

Equal variances assumed

1,349

23

0,190

0,04630

Equal variances not assumed

1,328

18,391

0,200

0,04630

STMT

Equal variances assumed

0,170

23

0,866

0,13333

Equal variances not assumed

0,155

13,473

0,879

0,13333

Simple condition

Equal variances assumed

0,667

23

0,511

0,04000

Equal variances not assumed

0,676

20,264

0,507

0,04000

Complex condition

Equal variances assumed

0,972

23

0,341

0,03125

Equal variances not assumed

1,014

21,944

0,322

0,03125

Table 8 shows t-tests of the variables of interest of the Music-educated group and SES.

Variables 

t

df

Mean Difference

Perceptual musical ability

Equal variances assumed

1,040

23

0,309

3,52206

Equal variances not assumed

1,014

12,977

0,329

3,52206

Perceptual language ability

Equal variances assumed

0,048

23

0,962

0,00196

Equal variances not assumed

0,049

14,726

0,962

0,00196

STMT

Equal variances assumed

-0,098

23

0,922

-0,09559

Equal variances not assumed

-0,088

10,836

0,931

-0,09559

Simple condition

Equal variances assumed

0,862

23

0,398

0,06912

Equal variances not assumed

0,937

17,102

0,362

0,06912

Complex condition

Equal variances assumed

-1,048

23

0,305

-0,03768

Equal variances not assumed

-0,989

12,019

0,342

-0,03768

Reviewer 2 Report

The manuscript entitled “Misjudgement of one’s own performance? Exploring Attention Deficit (Hyperactivity) Disorder (ADHD) and individual difference in complex music and foreign language perception” is a valuable and engaging read for those who are interested in understanding ADHD. The authors attempt to replicate the findings of an earlier study involving children and adolescents. In my modest opinion, the study merits publication after a few concerns are addressed. The following are such concerns:  

The abstract should be a one-paragraph summary of the research conducted by the authors, including purpose (including rationale), methodology, results, and implications or applications of the results.

The rationale of the study needs to be explained more broadly. Namely, the authors’ earlier findings were that ADHD children and adolescents displayed deficits in both processing of complex musical stimuli and speech perception of unfamiliar stimuli when compared to controls”. In their current study, why did the authors want to know whether these findings would be replicated in young adults? What are the brain areas of a young adult with ADHD that may be functionally different from those of a child or adolescent? What is the difference between an adolescent and a young adult in terms of brain functioning?

At the end of the introductory section, questions need to be accompanied by hypotheses. Each hypothesis also must be linked to a succinct rationale.

A clear operational definition must be given for the dimensions “complexity” and “unfamiliarity”. The two dimensions need to be clearly defined so that the reader can understand the deficits that the authors report in ADHD subjects.

 The ability to estimate accurately a stimulus may entail both a perceptual deficit and/or a metacognitive awareness difficulty. How do the authors distinguish between the two accounts? Can perceptual deficits merely imply difficulties in the estimation of one’s capabilities? Is there evidence that overestimation in ADHD encompasses tasks for which ADHD subjects do not display any deficits?

Are the samples of sufficient size for the analyses that the authors conduct on their data?

In the discussion section, the implications and applications of the current findings may need to be expanded. What do the current findings add to the extant literature on human development and specifically to human development in ADHD subjects?

The limitations of the current study may need to be discussed thoroughly.

Moderate editing of the English language is required.

Author Response

Open Review

Quality of English Language

( ) I am not qualified to assess the quality of English in this paper
( ) English very difficult to understand/incomprehensible
( ) Extensive editing of English language required
(x) Moderate editing of English language required
( ) Minor editing of English language required
( ) English language fine. No issues detected

Yes

Can be improved

Must be improved

Not applicable

Does the introduction provide sufficient background and include all relevant references?

(x)

( )

( )

( )

Are all the cited references relevant to the research?

(x)

( )

( )

( )

Is the research design appropriate?

(x)

( )

( )

( )

Are the methods adequately described?

(x)

( )

( )

( )

Are the results clearly presented?

( )

(x)

( )

( )

Are the conclusions supported by the results?

( )

(x)

( )

( )

Comments and Suggestions for Authors

The manuscript entitled “Misjudgement of one’s own performance? Exploring Attention Deficit (Hyperactivity) Disorder (ADHD) and individual difference in complex music and foreign language perception” is a valuable and engaging read for those who are interested in understanding ADHD. The authors attempt to replicate the findings of an earlier study involving children and adolescents. In my modest opinion, the study merits publication after a few concerns are addressed. The following are such concerns:  

Dear reviewer many thanks for your kind and valuable comments on our manuscript. We are very thankful for them and tried to integrate your kind suggestions into our manuscript and hope to have addressed your suggestions satisfactorily. We answered all your questions point by point and we also added the new passages here as well as we included the line number of the manuscript where we marked larger changed parts in grey. We hope that this saves you time.

  1. The abstract should be a one-paragraph summary of the research conducted by the authors, including purpose (including rationale), methodology, results, and implications or applications of the results.

Many thanks for this comment. We rewrote and shortened the abstract according to your suggestions and deleted around one third of the things we had previously and hope that this is better now.

See abstract in the manuscript and below

“In previous research we detected that children and adolescents who were diagnosed with ADHD showed deficits in both complex auditory processing of musical stimuli and in musical performance when compared to controls. In this study we were interested in whether we could detect similar or distinct findings when we use foreign speech perception.

Therefore, we recruited musically naïve participants (N=25), music-educated participants (N=25) and participants diagnosed with ADHD (N=25) who were assessed for their short-term memory (STM) capacity, the ability to discriminate music and speech stimuli, as well as we collected self-ratings of the participants’ music and language performances. As expected, we found that young adults with ADHD show deficits in the perception of complex music and difficult speech perception stimuli. We also found that STM capacity was not impaired in young adults with ADHD and may not persist into young adulthood. In addition, subjective self-estimation about the participants’ language and music performances revealed that the ADHD group overestimated their performance competence relatively compared to both control groups.

As a result, the findings of our study suggests that individuals diagnosed with ADHD require a special training program that does not only focus on improving performance in perceptual skills of music and language but also requires metacognitive training to develop realistic self-assessment skills.”

  1. The rationale of the study needs to be explained more broadly. Namely, the authors’ earlier findings were that ADHD children and adolescents displayed deficits in both processing of complex musical stimuli and speech perception of unfamiliar stimuli when compared to controls”. In their current study, why did the authors want to know whether these findings would be replicated in young adults? What are the brain areas of a young adult with ADHD that may be functionally different from those of a child or adolescent? What is the difference between an adolescent and a young adult in terms of brain functioning?

Many thanks for this comment. We apologize for this misunderstanding. We did not make it clear enough that we did not only replicate the study but also integrated new measures (language and short-term memory tasks) which was the main aim of the study. We made that more precise in the text and hope that the motivation of the study is now clear.

See lines 184-188 in the manuscript and below

“In this study we wanted to assess individual differences in the perception of music in the same way we did in previous research. In addition, we integrated two further measures that were found to be interrelated to musical abilities, namely speech perception and STM capacity measures. We recruited young adults possessing ADHD, as well as two control groups.”

  1. At the end of the introductory section, questions need to be accompanied by hypotheses. Each hypothesis also must be linked to a succinct rationale.

Thank you for this comment. We included hypotheses at the end of the introduction (see lines 191 ff in the manuscript) and linked them to a succinct rationale.

“First, we wanted to know whether young adults diagnosed with ADHD also show deficits in the ability to discriminate complex melodies compared to controls and to find out whether previous findings could be replicated when we test young adults. As found in previous research on children and adolescents (27, 26), we suggested that young adults with ADHD perform lower in complex musicality measures  (Q1).

Second, we wanted to assess whether individuals possessing ADHD also show lower performance in STM ability and the ability to discriminate unfamiliar foreign language material in simple and difficult conditions. We suggested that individuals with ADHD perform lower in the language tasks compared to the control groups since previous research on healthy adults (33, 92, 75) has shown that language ability and also enhanced STM capacity are intercorrelated with elaborate musical skills (Q2).

Third, we also wanted to assess whether the self-estimation variables of the language and music performances differed across groups since we expected individuals diagnosed with ADHD to overestimate their performance as often provided in previous studies (21, 24). Therefore, we also included a music-educated control group in the study since previous studies have shown that individuals with musical training assess their performances very well (75, 22, 23). This should help determine the degree to which young adults with ADHD misinterpret their own performance(Q3).”

  1. A clear operational definition must be given for the dimensions “complexity” and “unfamiliarity”. The two dimensions need to be clearly defined so that the reader can understand the deficits that the authors report in ADHD subjects.

Thank you for this comment. We defined the complex and unfamiliarity dimension to describe the deficits more precisely.

See  lines 114-117 in the manuscript and below:

“In the context of this study the term “complex” refers to difficult music discrimination tasks that consist of longer melodies which have to be remembered and to difficult language measures that consist of more than one language constituent which have to be remembered and discriminated. In this study the term “unfamiliarity” means that individuals do not speak, comprehend, or had learnt any of the languages before.”

  1. The ability to estimate accurately a stimulus may entail both a perceptual deficit and/or a metacognitive awareness difficulty. How do the authors distinguish between the two accounts?

Many thanks for this valuable comment. We integrated this also in the manuscript and addressed to rational idea of our research design in more detail.

Referring to perceptual deficit, we decided to use a design in which we had language tasks which were simple and more difficult. We suggested that if individuals with supposed deficits perform differently in these conditions, we can suggest having found a perceptual issue. This indeed we found for musicality measures in previous research. Therefore, we decided to have this approach for the language tasks in this study as well. We included a short statement in the discussion.

See lines 536-542 in the manuscript and below:

“We suggested that if individuals with ADHD perform differently in simple and complex conditions, we can suggest having found a perceptual issue. Indeed, we found this when we analyzed both language conditions separately. The separate ANOVAs of the simple and difficult language perception tasks showed that group differences between the two control groups and the ADHD group were only observed when the language tasks were more difficult, while in the simple condition no statistically significant difference across all three groups was detected (see section 3.3).”

And as you well formulated, how do we distinguish between metacognitive awareness difficulty? We followed here two approaches. First, one suggestion of other researchers is that adults with ADHD are less aware of their own cognitive deficits and therefore have a tendency to overestimate their cognitive performance relative to controls (see lines 201 ff in the manuscript), and the second previous findings of our own research. To the second first, we decided to have two control groups to be integrated into the research design because we expected performance differences for the music-educated group to both other groups in the language and music tasks. In addition, based on previous research we knew that music-educated and non-musicians can estimate their own music and language abilities quite well as they are intercorrelated. This then would include that the music-educated group will also provide higher self-estimation scores compared to the musically naïve group. Consequently, we suggested that if the self-estimation scores of the ADHD do show remarkable differences from both control groups relative to the performance scores, we can also talk about misconceptions of the own abilities. We therefore included a short passage in the introduction since we thought that here fits best.

See lines 201-210 in the manuscript and below:

“Third, we also wanted to assess whether the self-estimation variables of the language and music performances differed across groups since we expected individuals diag-nosed with ADHD to overestimate their performance as often provided in previous studies (21, 24). Therefore, we also included a music-educated control group in the study since previous studies have shown that individuals with musical training assess their performances very well (75, 22, 23). We suggested that if the self-estimation scores of the participants with ADHD show remarkable differences from both control groups relative to the performance scores, it can be suggested that This should help de-termine the degree to which young adults with ADHD misinterpret or overestimate their own performance(Q3).).”

  1. Is there evidence that overestimation in ADHD encompasses tasks for which ADHD subjects do not display any deficits?

Many thanks for this interesting remark. With our research we cannot answer this question at this stage, but we think this is a very interesting idea which should be studied in more detail in future. Therefore, we included this short passage in the manuscript in the discussion section.

See lines 623-625 in the manuscript and below:

“In addition, future studies should also address whether the overrating of performances in individuals with ADHD also encompasses tasks for which they do not show any deficits.”

  1. Are the samples of sufficient size for the analyses that the authors conduct on their data?

Thank you for this comment. To perform Manova and a follow up analysis discriminant analysis, as an absolute minimum, there must be at least as many cases in each group of the independent variable as there are dependent variables. This is indeed clearly given. However, we agree that the more participants are included the more robust the result. Therefore, we also include this in the limitations of the study where we say that the results should be replicated decided for this type of analysis.

 See lines 620-622 in the manuscript and below

“Future studies have to consider larger sample size and include different tasks which assess various domains in language and music to understand the impact of ADHD more precisely. “

  1. In the discussion section, the implications and applications of the current findings may need to be expanded. What do the current findings add to the extant literature on human development and specifically to human development in ADHD subjects?

Many thanks this is what also reviewer 1 wanted to know and we included a short section in the conclusion.

See lines 638-648 in the manuscript and below

“As a result, specialized music and language learning programs for individuals diagnosed with ADHD should consider different domains.  Based on our findings teaching methods for individuals with ADHD should focus on how difficult and complex music and language tasks could be presented in a simple way. In addition, metacognitive training should be included that involves aspects to strengthen the self-esteem and self-image and facilitates the development of realistic self-assessment. In this respect a musical training program would be highly beneficial for many reasons. First, musical training improves perceptual skills in both domains, music and language. Second, in music education self-assessment of the own musical abilities is an important part of musical training. Third, musical training increases the self-confidence and self-esteem (110).”

  1. The limitations of the current study may need to be discussed thoroughly.

Many thanks for this comment we included limitations in the manuscript.  

See lines 618-624 in the manuscript and below:

“This study also has limitations. The main aim of the study was to look at music and language perception ability. This means that this study only provides limited insights in deficits of foreign language capacity and musical skills of individuals with ADHD. Future studies have to consider larger sample size and include different tasks which assess various domains in language and music to understand the impact of ADHD more precisely. In addition, future studies should also address whether the overrating of performances in individuals with ADHD also encompasses tasks for which they do not show any deficits. “

Round 2

Reviewer 2 Report

The authors have considerably improved the content of their manuscript. In my modest opinion, it is now an engaging and valuable contribution to the literature on attention deficit hyperactivity disorder.

Minor editing of the English language is required.